# Role of Internal Stress in the Early-Stage Nucleation of Amorphous Calcium Carbonate Gels

**Qi Zhou [1], Tao Du [2], Lijie Guo [3,*] , Gaurav Sant [4] and Mathieu Bauchy [5,*]**

[1]  Physics of AmoRphous and Inorganic Solids Laboratory (PARISlab), University of California, Los Angeles, CA 90095-1593, USA; qi1197@ucla.edu

[2]  Key Lab of Structures Dynamic Behavior and Control (Harbin Institute of Technology), Ministry of Education, Harbin 150090, China; dutaohit@gmail.com

[3]  BGRIMM Technology Group, Beijing 100160, China

[4]  Laboratory for the Chemistry of Construction Materials (LC2), California Nanosystems Institute (CNSI), Institute for Carbon Management (ICM), Department of Civil and Environmental Engineering, University of California, Los Angeles, CA 90095, USA; gsant@ucla.edu

[5]  Physics of AmoRphous and Inorganic Solids Laboratory (PARISlab), California Nanosystems Institute (CNSI), Institute for Carbon Management (ICM), University of California, Los Angeles, CA 90095-1593, USA

*  Correspondence: guolijie@bgrimm.com (L.G.); bauchy@ucla.edu (M.B.)



**Featured Application:** **This work offers new insights into the atomic-scale mechanism governing the early-stage nucleation of amorphous calcium carbonate gels, which is the key to the acceleration of the development of new methods, enabling low-cost $CO_2$ utilization by mineralization.**

**Abstract:** Although calcium carbonate ($CaCO_3$) precipitation plays an important role in nature, its mechanism remains only partially understood. Further understanding the atomic driving force behind the $CaCO_3$ precipitation could be key to facilitate the capture, immobilization, and utilization of $CO_2$ by mineralization. Here, based on molecular dynamics simulations, we investigate the mechanism of the early-stage nucleation of an amorphous calcium carbonate gel. We show that the gelation reaction manifests itself by the formation of some calcium carbonate clusters that grow over time. Interestingly, we demonstrate that the gelation reaction is driven by the existence of some competing local molecular stresses within the Ca and C precursors, which progressively get released upon gelation. This internal molecular stress is found to originate from the significantly different local coordination environments exhibited by Ca and C atoms. These results highlight the key role played by the local stress acting within the atomic network in governing gelation reactions.

**Keywords:** calcium carbonate; molecular dynamics; carbon utilization; gelation

## 1. Introduction

Calcium carbonate ($CaCO_3$) is ubiquitous in nature and plays an important role in biomineralization [1]. For instance, $CaCO_3$ binding phases are commonly formed by organisms that produce an exoskeleton, e.g., snails, clams, and mollusks, and in cemented granular soils formed by bacteria [2,3]. On the technological side, calcium carbonate is key for $CO_2$ utilization approaches relying on mineralization [4], which offer a promising route to turn $CO_2$ (usually considered as a waste) into a resource, e.g., concrete binders [5,6].

The development of novel carbonation routes enabling low-cost and low-energy $CO_2$ capture has thus far been largely limited by a lack of understanding of the physical and chemical features

that govern the carbonation process [4]. Indeed, despite its ubiquitous nature, the mechanism of the precipitation of calcium carbonate remains poorly understood [7]. The carbonation process usually occurs via a dissolution-precipitation reaction, wherein a solid phase (e.g., $Ca(OH)_2$) dissolves in an aqueous environment, reacts with dissolved carbonate species, and reprecipitates as an amorphous, hydrated precursor calcium carbonate gel phase [8]. Crystalline $CaCO_3$ then forms upon the drying and crystallization of the amorphous precursor [9]. Notably, the nucleation of amorphous calcium carbonate is barrierless in highly supersaturated calcium carbonate solutions and does not follow conventional nucleation pathways [10]. Although the existence of some stable prenucleation clusters has been suggested, the atomic scale mechanism of calcium carbonate nucleation in an aqueous condition remains largely unknown [11,12]. Since amorphous calcium carbonate often acts as a precursor before the formation of subsequent crystalline carbonate phases, it is important to understand the formation mechanism of the amorphous calcium carbonate gel [13,14].

As an alternative route to experiments, molecular dynamics (MD) simulations can offer some valuable insights into the atomic mechanism governing the precipitation of gels [15,16]. Indeed, although MD simulations are limited to small systems and timescales, they can offer a direct access to the time-dependent structure of calcium carbonate during its early-stage nucleation [17]. In particular, Raiteri et al. recently developed a thermodynamically-consistent forcefield, allowing the full modeling of the nucleation of carbonates, ranging from aqueous complexes to solid carbonates [18,19]. This forcefield was found to accurately reproduce the dynamics and thermodynamics of alkaline-earth carbonate solutions and to yield an excellent agreement with experimental data. These recent developments now make it possible to investigate the atomic mechanism behind the carbonation reaction.

Here, based on MD simulations, we investigate the early-stage nucleation of an amorphous calcium carbonate gel in supersaturated conditions. We show that the gelation reaction manifests itself by the formation of some calcium carbonate clusters that grow over time. As a key novelty resulting from this study, we demonstrate the existence of some local, competing atomic stress acting in calcium carbonate gels, which arises from the significantly different coordination environments exhibited by Ca and C atoms. Such internal stress is found to be progressively released upon gelation and acts as a driving force for early-stage nucleation.

## 2. Methods

### 2.1. Simulation Methodology

The calcium carbonate gelation simulations are conducted based on the method presented by Cormack et al. and described in the following [20]. A $CaCO_3$ hydrated gel is simulated with the large-scale atomic/molecular massively parallel simulation (LAMMPS) package [21]. The system comprises 11,840 atoms with a $CaCO_3/H_2O$ molar ratio of 1/60, which is highly supersaturated. Although this concentration might not mimic realistic solution conditions, such a high concentration allows us to accelerate the gelation reaction. First, $Ca^{2+}$ ions and $CO_3^{2-}$ and $H_2O$ molecules are randomly placed in a cubic box of 55 Å in length, with periodic boundary conditions, while ensuring the absence of any unrealistic overlap between atoms using the Packmol package [22]. In the following, Ow and Oc refer to the O atoms that belong to $H_2O$ and $CO_3^{2-}$ molecules, respectively. A snapshot of initial configuration is shown in Figure 1a. The system is first relaxed at zero pressure and 300 K for 25 ps in the isothermal-isobaric (*NPT*) ensemble, using a Nosé-Hoover thermostat and the barostat developed by Melchionma et al. [23,24] This duration is found to be long enough to ensure a convergence of pressure and volume. The early-stage gelation dynamics of the system are then studied by conducting a 10 ns MD run in the *NVT* ensemble. The velocity-Verlet integration algorithm is employed for the description of the atomic motion, with a time step of 0.25 fs. For all the simulations, we adopt the forcefield parameterized by Raiteri et al., which models the atoms as rigid ions with fixed charges [18]. Although more complex forcefields are available (e.g., ReaxFF [25,26] or fluctuating charge model [27]), they come with a significantly increased computational cost, which would not allow us to simulate

such a large system over such an extended timescale. In addition, the forcefield developed by Raiteri et al. presents the advantage of offering an excellent description of the dynamics and thermodynamics of carbonates—both in solution and as solids [18]. The full parameterization of this forcefield can be found in Ref. [18]. For illustration purposes, Figure 1b,c show some snapshots of the simulated configuration after 0, 5, and 10 ns of dynamics.

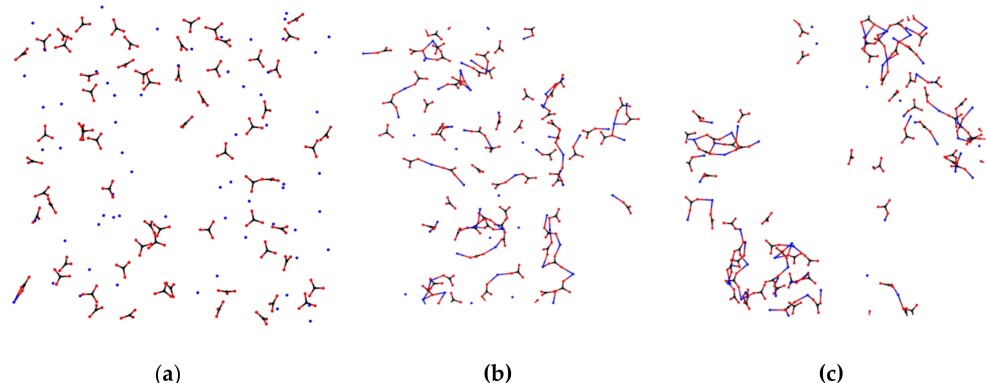

| (a) | (b) | (c) |

**Figure 1.** Snapshots of the simulated amorphous calcium carbonate system after (**a**) 0, (**b**) 5, and (**c**) 10 ns of gelation. Carbon, oxygen, and calcium atoms are indicated in black, red, and blue, respectively. Water molecules are here omitted for clarity.

## 2.2. Structural Analysis

To describe the time evolution of the atomic topology of the amorphous calcium carbonate gel, we compute the coordination number of each atom by enumerating the number of neighbors present in its first coordination shell. The distance cutoffs are determined for each atom as the first minimum distance of the partial pair distribution functions (i.e., 1.5 and 3.0 Å for C–O and Ca–O bonds, respectively). Based on the identification of each C–O and Ca–O bond, we then compute the size and number of carbonate clusters, wherein a cluster is defined as a group of interconnected C, Ca, and O atoms. The size of each cluster is then given by the number of C atoms belonging to the cluster. The clustering analysis is conducted by using the OVITO software [28].

## 2.3. Computation of the Local Atomic Stress

To assess the existence of some local instabilities within the atomic network, we adopt the concept of "stress per atom" introduced by Egami [29,30]. Note that, strictly speaking, stress is intrinsically a macroscopic property and, hence, is ill-defined for individual atoms [31]. Nevertheless, an atomic-level stress tensor $\sigma_i^{\alpha\beta}$ can be defined for each atom $i$ by expressing the contribution of each atom to the virial of the system [32]:

$$\sigma_i^{\alpha\beta} = \frac{1}{V_i}\Sigma_j r_{ij}^\alpha \cdot F_{ij}^\beta, \tag{1}$$

where $V_i$ is the volume of atom $i$, $r_{ij}$ the distance between atoms $i$ and $j$, $F_{ij}$ the interatomic force applied by atom $j$ on atom $i$, and the indexes $\alpha$ and $\beta$ refer to the projections of these vectors along the Cartesian directions $x$, $y$, or $z$. The volume $V_i$ of each atom is defined as the Voronoi volume. By convention, a positive stress represents here a local state of tension, whereas a negative one represents a local state of compression. This approach was recently used to quantify the internal stress exhibited by stressed–rigid atomic networks [33] and mixed alkali glasses [32,34–36]. It should be noted that, in the thermodynamic sense, stress is only properly defined for a large ensemble of atoms, so that the physical meaning of the "stress per atom" is unclear. Nevertheless, this quantity can conveniently capture the existence of local instabilities within the gel, due to competitive interatomic forces [37].

## 3. Results

We first focus on the connectivity of the Ca atoms upon gelation. Figure 2 shows the evolution of the average number of Ca–O bonds per Ca atom over time. We find that, overall, the average coordination number of the Ca atoms remains fairly constant over time throughout the gelation process, namely, around 6.5. However, we observe that the type of O atoms belonging to the first coordination shell of Ca atoms changes over time. As expected, Ca atoms are initially entirely surrounded by Ow atoms, that is, O atoms belonging to water molecules. At this point, $Ca^{2+}$ ions act as isolated hydrated cations. As gelation proceeds, we observe that Ca–Ow bonds are gradually replaced by Ca–Oc, that is, Ca atoms gradually start to form some bonds, with some O atoms belonging to $CO_3^{2-}$ molecules. This signals a gradual increase in the degree of polymerization of the calcium carbonate gel (see Figure 1).

The formation of Ca–Oc–C bonds (see Figure 2) results in the appearance of carbonate clusters, that is, some groups of C atoms that are interconnected via Ca atoms (i.e., C–Oc–Ca–Oc–C bonds). Figure 3a shows the number of carbonate clusters as a function of time. As expected, the system initially comprises a large number of small isolated clusters. As gelation proceeds, the increase in the number of bonds between $CO_3^{2-}$ groups results in a decrease in the number of carbonate clusters, which is concurrent with an increase in their average size (see Figure 1). As shown in Figure 3b,c, both the size of the largest carbonate cluster (as described by the number of C atoms it comprises) and the average size of the carbonate clusters quickly increase over time. This suggests that the carbonate gelation process is percolative in nature.

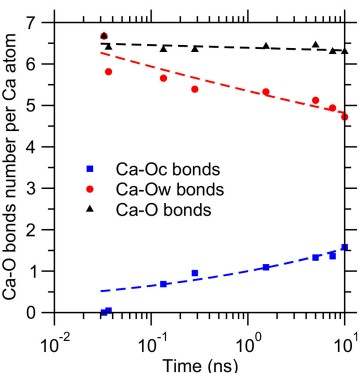

**Figure 2.** Evolution of the average number of Ca–O bonds per Ca atom as a function of time. The data include the contributions of Oc (i.e., O connected to a C atom) and Ow atoms (i.e., O belonging to a water molecule). The lines are to guide the eye.

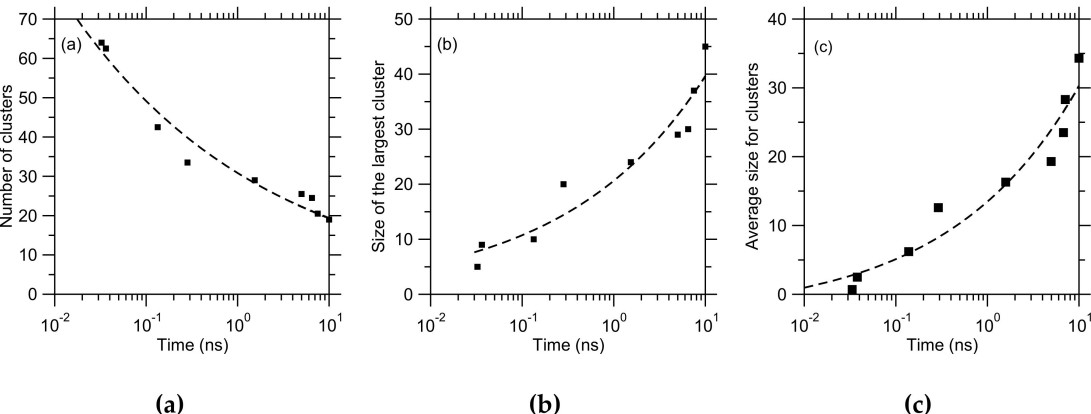

**Figure 3.** (**a**) Number of carbonate clusters, (**b**) size of the largest cluster (expressed in terms of the number of C atoms), and (**c**) average size of the carbonate clusters (expressed in terms of the number of C atoms), as a function of time during the early-stage nucleation of the amorphous calcium carbonate gel. The lines are to guide the eye.

## 4. Discussion

We now discuss the local stability of the structure by computing the magnitude of the local stress undergone by each atom in the structure. Figure 4 shows the evolution of the local stress undergone by Ca, Ow, C, and Oc atoms upon gelation. We first focus on the state of stress of the Ca atoms. We observe that, initially, Ca atoms are experiencing a local state of tension (i.e., positive stress, see Figure 4a). On the other hand, Ow atoms are under compression (i.e., negative stress, see Figure 4b). The origin of this state of stress can be understood in terms of the number of neighboring water molecules around Ca atoms. In detail, we find that the average O–O distance around Ca cations is 2.72 Å, which is smaller than the average O–O distance in bulk water (2.80 Å) [38]. Such O–O distances match with experimental data [11]. This indicates that, by being attracted by the central Ca cation, neighboring water molecules tend to get closer to each other than they are in bulk water. As a result, the neighboring water molecules enter a state of local compression (since they repulse each other)—as evidenced in Figure 4b. In turn, the mutual repulsion between water molecules tends to stretch the central Ca atom, so that Ca atoms experience a state of tension—as evidenced in Figure 4a. These local tensile and compressive forces mutually compensate each other, so that the $CaO_6$ polytope is overall at a (metastable) mechanical equilibrium. This behavior is schematically illustrated in Figure 5a. The state of tension of Ca atoms echoes the case of silicate glasses, wherein Si atoms also undergo a local tensile stress [33]. Nevertheless, we note that it remains whether this feature (e.g., the lowering of the O–O distance around Ca atoms) is generic or specific to this system [11,39,40]. It should also be noted that the coordination number of Ca atoms may depend on the solution pH and Ca concentration [41], which, in turn, could affect their state of stress. Clearly, the effect of the solution chemistry on the state of stress of the gel precursors would deserve follow-up investigations.

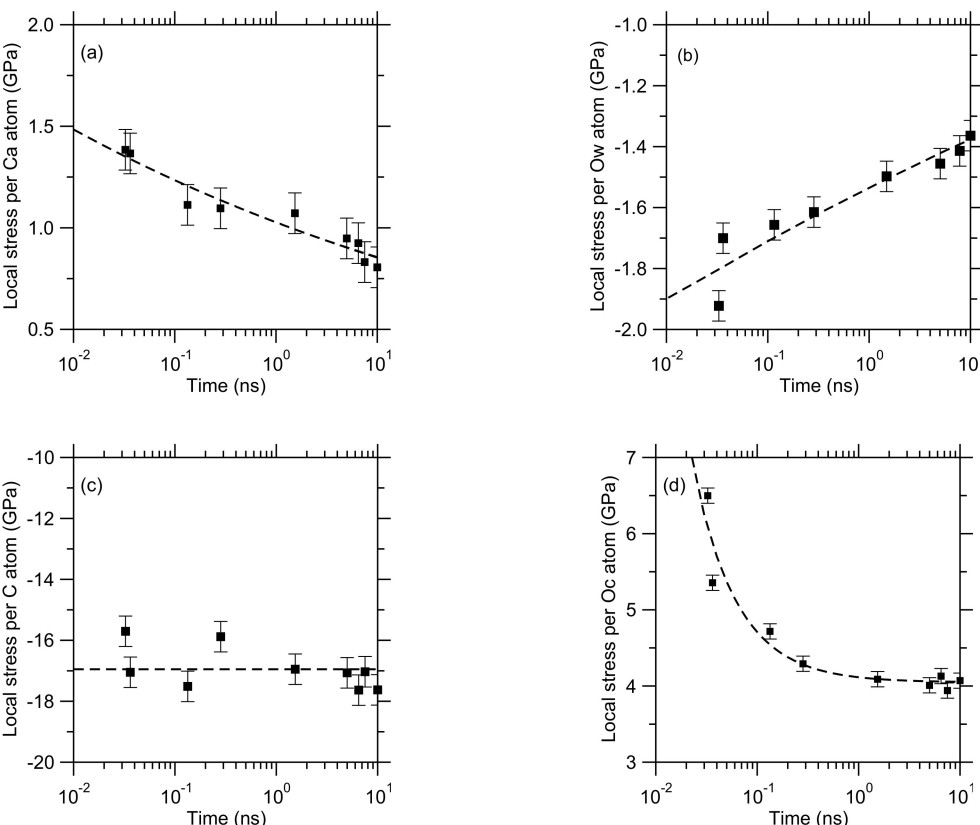

**Figure 4.** Evolution of the local stress experienced by (**a**) Ca, (**b**) Ow, (**c**) C, and (**d**) Oc atoms as a function of time. Positive and negative stress values indicate a state of local tension and compression, respectively.

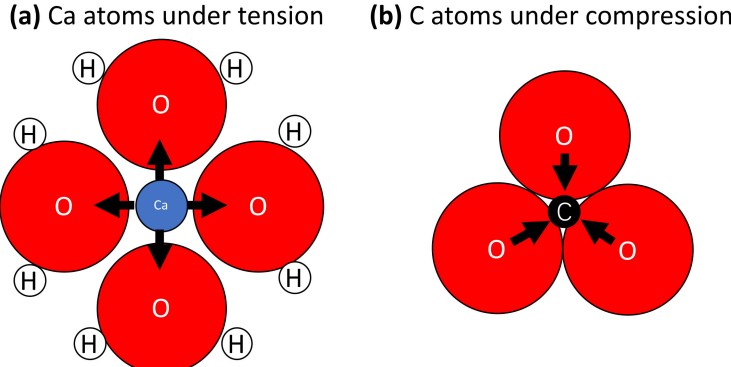

**Figure 5.** Schematic description of the origin of the (**a**) tensile and (**b**) compressive stress experienced by Ca and C atoms, respectively. Note that 6-fold coordinated Ca atoms also exhibit 2 additional O atoms above and below this plane. The circles are intended to represent the radii of the atoms (which are not at scale to enhance readability).

In contrast, C and Oc present a state of compression and tension, respectively (see Figure 4c,d), which contrasts with the case of Ca and Ow atoms. This can be understood from the fact that, in contrast to Ca, C atoms have a low coordination number (around 3), so that their first coordination shell exhibits a low density of Oc atoms. As such, in contrast with the case of the water molecules around Ca cations, no significant mutual coulombic repulsion among O atoms is observed around C atoms. In fact, we find that the average C–O distance around C atoms is 1.30 Å (in agreement with experimental data [11]), which is notably shorter than the equilibrium C–O distance of 1.56 Å (based on the sum of their ionic radii) [42]. Hence, the fact that C atoms are locally under compression (as evidenced by Figure 4c) arises from the fact that they are surrounded by very close O neighbors, which forces their effective radius to be smaller than their equilibrium ionic radius. In turn, the Oc neighbors around the central C atom undergo a state of tension (as evidenced by Figure 4d) for the $CO_3$ polytope to be at an overall (metastable) mechanical equilibrium. This behavior is schematically illustrated in Figure 5b.

These results highlight the fact that, although the entire gel is at zero pressure, the network locally exhibits some localized compressive and tensile stress at the atomic level, which arises from the significantly different coordination numbers of C and Ca atoms. However, as shown in Figure 4, such internal stress gradually gets released as the gelation proceeds. This suggests that the local stress that is initially present in the system acts as a driving force that drives the gelation transformation. This can be understood as follows. The existence of local stress around C and Ca atoms comes with an elastic energy penalty. As the gelation reaction proceeds, the formation of linkages among Ca and C atoms—which initially experience opposite states of stress—reduces the magnitude of such stress. This arises from the gradual replacement of Ow by Oc atoms in the first coordination shell of the Ca atoms, so that O atoms initially experiencing some states of compression and tension are mutually able to release each other's stress. Eventually, we find that such a combination of atomic species initially experiencing opposite states of stress mostly benefits Ca atoms—which experience a significant drop in their internal stress—whereas the stress experienced by C atoms remains largely unaffected. This likely arises from the difference in stiffness between Ca–O and C–O bonds.

We now investigate how the release of the local internal stress upon gelation manifests itself in the structure of the gel. To this end, we first compute the evolution of the distribution of the O–O neighbor-neighbor distances around Ca and C atoms (see Figure 6). We first note that, in both cases, the distributions gradually become sharper, which echoes the fact that the gel becomes more and more stable. Importantly, we find that the average O–O distance around Ca cations tends to slightly increase over time (see Figure 6a). This is consistent with the fact that the compressive stress experienced by the O atoms around Ca cations tends to decrease over time—since water molecules gradually get further away from each other and overlap less with each other. In contrast, we note that the average O–O

distance around C atoms tends to slightly decrease over time (see Figure 6b). This is consistent with the fact that the tensile stress experienced by the O atoms around C atoms tends to decrease over time.

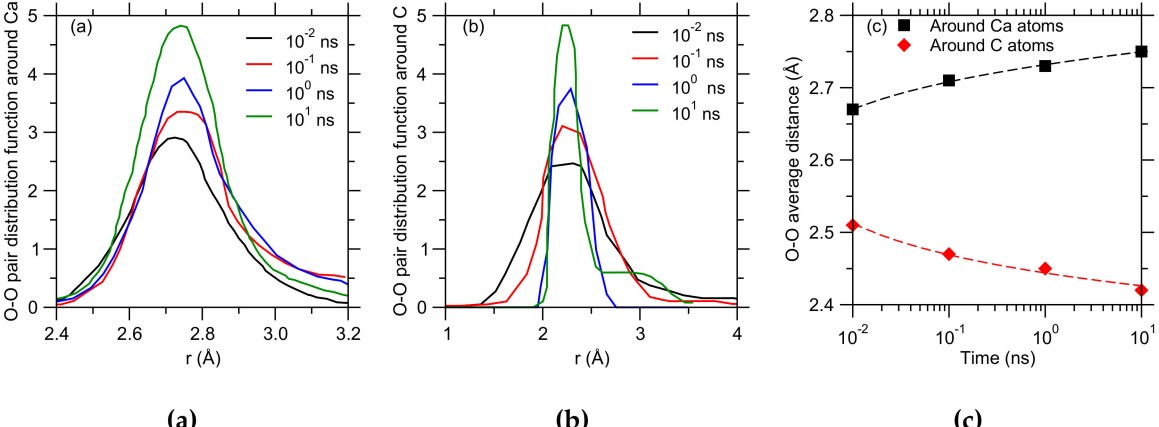

**Figure 6.** Distribution of the O–O distances around (**a**) Ca and (**b**) C atoms, computed at select times during the polymerization of the calcium carbonate gel. (**c**) Average O–O distance around Ca and C atoms as a function of time. The lines are to guide the eye.

Finally, we focus on the structural manifestation of the release of the local tensile stress experienced by Ca atoms. To this end, we compute the evolution of the Ca–Oc and Ca–Ow partial pair distribution functions (see Figure 7). The average Ca–O distance obtained herein agrees with previous experimental data [11,39,40]. We observe that the average Ca–Ow bond length tends to decrease over time, while the average Ca–Oc bond length tends to increase (see Figure 7c). Eventually, the average Ca–Oc bond length (2.256 Å) becomes notably larger than the average Ca–Ow bond length (2.242 Å), which indicates that the $CaO_6$ polytope becomes less spherical and symmetric (i.e., since different Ca–O bonds exhibit different lengths). The decoupling between Ca–Ow and Ca–Oc bond lengths is at the origin of the increase in the average O–O distance around Ca atoms (see Figure 6c). Overall, due to the predominance of Ca–Ow bonds, the average Ca–O bond length tends to decrease over time (see Figure 7c). The decrease in the Ca–O distance indicates that the central Ca atoms become less and less stretched over time. This is consistent with the fact that the magnitude of the local tensile stress tends to decrease over time. Overall, these results highlight the fact that the formation of calcium carbonation clusters (i.e., the formation of Ca–O–C linkages through the replacement of Ca–Ow by Ca–Oc bonds) directly results in the lowering of the local stress exhibited by Ca atoms. This establishes gelation as an efficient mechanism to release the internal stress that is initially present within the precursors. A similar mechanism was also suggested to occur in hydrated calcium aluminosilicate gels [43], wherein the formation of Si–O–Al linkages was also noted to result in a decrease in the magnitude of the internal stress exhibited by Si (experiencing local tension) and Al (experiencing local compression) precursors. This suggests that the relationship between local internal stress and gelation might apply to a broad array of hydrated systems [15,43].

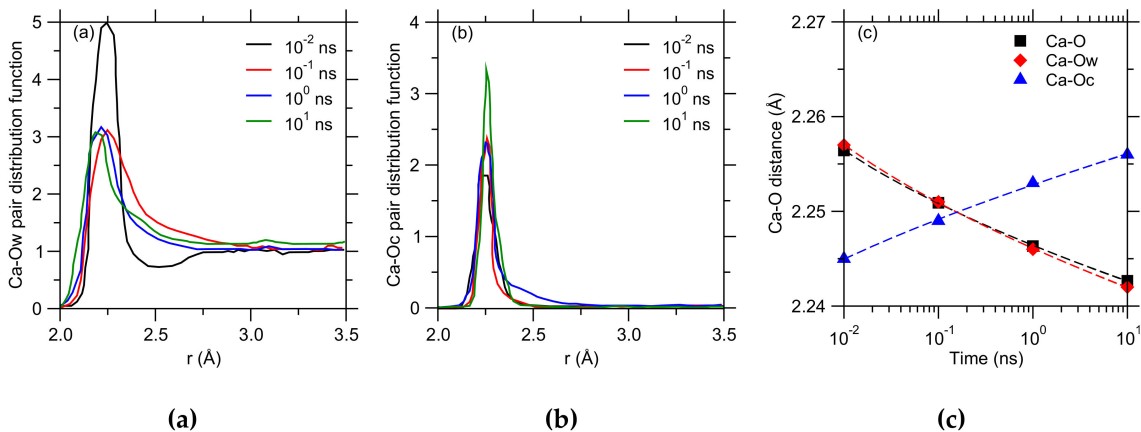

**Figure 7.** (**a**) Ca–Ow and (**b**) Ca–Oc partial pair distribution functions computed at select times during the polymerization of the calcium carbonate gel. (**c**) Average Ca–O, Ca–Ow, and Ca–Oc bond length as a function of time. The lines are to guide the eye.

## 5. Conclusions

Overall, these results highlight the role played by the internal stress in governing the early-stage nucleation of amorphous calcium carbonate. We find that, although the system is, overall, at zero pressure, the $CaO_6$ and $CO_3$ polytope precursors initially experience some competing internal stress, which arises from their significantly different coordination states. Such stress acts as an elastic energy penalty that thermodynamically promotes the polymerization of the gel—so that the combination of polytopes undergoing some opposite states of stress results in an overall decrease in the magnitude of the local stress acting in the atomic network. These results highlight the key role played by local atomic instabilities in the precursor species in driving sol–gel reactions at the atomic level.

**Author Contributions:** Conceptualization, G.L., G.S. and M.B.; Formal analysis, Q.Z.; Funding acquisition, M.B.; Investigation, Q.Z.; Methodology, T.D.; Project administration, M.B.; Resources, G.L.; Software, Q.Z. and T.D.; Supervision, M.B.; Visualization, Q.Z.; Writing—original draft, Q.Z.; Writing—review and editing, Q.Z., T.D., G.L., G.S. and M.B. All authors have read and agreed to the published version of the manuscript.

**Funding:** This research was supported by the Beijing General Research Institute of Mining & Metallurgy (BGRIMM) Technology Group and the U.S. National Science Foundation (DMREF: 1922167).

**Conflicts of Interest:** The authors declare no conflict of interest. The funders had no role in the design of the study; in the collection, analyses, or interpretation of data; in the writing of the manuscript, or in the decision to publish the results.

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
