# Peer review of "Role of Internal Stress in the Early-Stage Nucleation of Amorphous Calcium Carbonate Gels"

_applsci, doi:10.3390/app10124359_

Round 1

Reviewer 1 Report

The article is well presented and it describes the mechanism of the early stage nucletaion of an amorphous calcium carbonate gel.

Authors should replaced Figure 4 b beacuse it is not clear.

Authors should avoid lumping references.

Reviewer 2 Report

This report is very interesting as it pointed out the role of internal stress in the early stage nucleation of amorphous calcium

carbonate gels.  There is no doubt that internal stress should be one of the major driving forces to initial nucleation. This report showed clearly that the change of internal stress is one of the major driving forces to initial nucleation. I am glad to see that clearly.

Authors demonstrated the change of the internal stress but I'd like to suggest they show the change of the average cluster

composition in Figure. 3, which will show cleary the details of initial nucleation stage.

Reviewer 3 Report

Review: Role of Internal Stress in the Early-Stage Nucleation of Amorphous Calcium Carbonate Gels, by Zhou et al.
In the present manuscript, the authors performed molecular dynamics simulation to analyze the formation process of amorphous calcium carbonate. The idea to focus on the local stress could be interesting, however, from the reviewer’s point of view, the discussion described in this manuscript seems to be insufficient and should be required to further verification. My concerns are as follows:
The authors insisted that the driving force for the formation and aggregation of CaCO3 clusters are local stress, which is caused by the difference between coordination numbers of Ca (around 6) and C (around 3). However, generally speaking, the coordination number is determined by the ratio of the ionic radii between anion and cation, therefore I don’ believe the difference between local stresses of Ca (tensile) and C (compressive) can be explained only by the difference of coordination numbers. Figure 5 seems to be incorrect, because O atoms should be arranged around Ca three dimensionally while O atoms place within the same plane. In this case, O-O distances in the CaO6 units are usually longer than these in CO3 triangles. The authors should check based on their calculation results weather the density of O in the first coordination shells of Ca is higher than that in CO3 triangles.
Related to this point, the local atomic arrangement obtained in the present calculation was not considered in their discussion. How do the distances of O-O and Ca-O change during the calculation, and how do these changes affect the local stress?
Additionally, the effects of Ow and Oc on the local stress for Ca should be different because Oc are considered to be applied the tension in the direction to C atom. Although the authors pointed out Ca-Ow bonds are gradually replaced by Ca-Oc as the calculation proceeds, they did not consider the effect of this phenomenon on the local stress around Ca atoms. Furthermore, since the local stresses for Ca and Ow monotonically change throughout the present calculation (Figure 4) and O atoms in the CaO6 units are constantly replaced (Figure 2), it should be difficult to conclude CaO6 polytope is at equilibrium.
Although there have been many experimental and computational studies on the formation and structure of amorphous calcium carbonate, the authors did not properly refer these studies. How do the present results relate to the previous researches? For example, some previous studies have reported there is the case when coordination number of Ca is different from 6 in aqueous solution, amorphous, and crystalline phases. Is the previous result consistent with these works?
Overall, I think the authors should reconstruct their discussion based on the detail analysis of local atomic arrangement obtained by the present calculation, taking into consideration previous computational and experimental studies, before being considered for publication.

Round 2

Reviewer 3 Report

I think that the authors made the proper effort to improve the manuscript as referees suggested, however, I still have some concerns on the part added in the revised manuscript as listed below. I would like to recommend publishing this work after clarified on these points.

1. Page 5, lines 160-161.
I don’t believe that small O-O distance around Ca originated from the fact that Ca atoms have a large coordination number. This authors’ claim seems to be inconsistent with the general consideration that a larger ion has larger coordination number, and also with the present result that O-O distance around C with low coordination number is much smaller than that around Ca. Furthermore, the authors insisted that average O-O distance around a solute ion (Ca) is smaller than that in bulk water (lines 158-159). However, this feature should be observed also in the other systems, and it is difficult to explain the specificity of this system by using this phenomenon. The authors should reconsider discussion on this part.

2. Page 6-7, lines 212-227.
The authors mentioned that, as the calculation proceeds, O-O distance in CaO6 polytope becomes to be larger, while Ca-O in that polytope is smaller. However, it seems to be curious, and only from this information it should be difficult to imagine how the geometry of CaO6 changes. Does CaO6 polytope expand or shrink? The authors should add the detail explanation on this point.

3. Figure 7
Values in the vertical axis in Figure 7b are much larger than these in Figure 7a and 7c. If the authors treated the data in the different way, they should explain in detail.
